# Treatment and control of blood pressure in Welsh patients with and without depression: A study of whole-population electronic health records

Elizabeth A. Ellins[1]*, Richard Summers[1], Carla White[1], Ann John[1],
David P.J. Osborn[2], Keith Lloyd[1], Ashley Akbari[1], Michael B. Gravenor[1],
Julian P. Halcox[1]

1 Faculty of Medicine, Swansea University Medical School, Health and Life Science, Swansea University, Swansea, United Kingdom, 2 Division of Psychiatry, University College London, London, United Kingdom

* e.a.ellins@swansea.ac.uk

## Abstract

### Background

To explore differences in blood pressure treatment and control in patients with and without depression and a diagnosis of hypertension (HTN). Also to examine the possible impact of sex, socio-economic status (deprivation) and location of residence on any differences.

### Methods

A retrospective observational cohort study (2010–2019) using individual level linked anonymised routinely-collected electronic health record (EHR) data sources was carried out. Patients with a prior or new diagnosis of hypertension, with and without depression were included. Outcome variables were prescription of antihypertensive therapy (AHT) within one year of entering the study for prior HTN or post for new HTN and documentation of blood pressure <140/90 mmHg. Logistic regression was used to explore the association between depression and outcome variables adjusting for sex, age group, deprivation, location of residence and other risk factors.

### Results

Depression was associated with higher likelihood of AHT prescription in both prior (OR 1.71 95%CI 1.64–1.78 p<0.001) and new HTN patients (OR 2.67 95%CI 2.38–3.00 p<0.001). Similarly, depression was associated with successful blood pressure control in both prior (OR 1.42 95%CI 1.37–1.46 p<0.001) and new HTN (OR 1.23 95%CI 1.08–1.40 p<0.001). Females were less likely to be prescribed AHT than males, mainly driven by non-depressed females in both HTN groups. Depressed females were the most likely to have controlled blood pressure in both HTN groups.

**Data availability statement:** The data used in this study are available in the SAIL Databank at Swansea University, Swansea, UK, but as restrictions apply, they are not publicly available. All proposals to use SAIL data are subject to review by an independent Information Governance Review Panel (IGRP). Before any data can be accessed, approval must be given by the IGRP. The IGRP carefully considers each project to ensure the proper and appropriate use of SAIL data. When access has been granted, it is gained through a privacy-protecting trusted research environment (TRE) and remote access system referred to as the SAIL Gateway. SAIL has established an application process to be followed by anyone who would like to access data via SAIL at https://saildatabank.com/data/apply-to-work-with-the-data/.

**Funding:** EE, JH, AJ. KL, DO, AA & MG British Heart Foundation project grant PG/21/10631 https://www.bhf.org.uk/for-professionals/information-for-researchers Funders did not play any role in the study design, data collection and analysis, decision to publish, or preparation of the manuscript.

**Competing interests:** I have read the journal's policy and the authors of this manuscript have the following competing interests: JPH, AA and MBG are investigators on an unrestricted research grant from Amgen Inc., related to cardiovascular risk management in clinical practice, which is investigating related issues but does not overlap in scope with the current work.

## Conclusion

Patients with depression are more likely to be prescribed AHT and have documented blood pressure control. Sex differences existed in treatment and control, indicating opportunities for potential improvements in these areas.

## Introduction

Hypertension (HTN) is a major risk factor in the development of cardiovascular disease [1]. Therefore, management and control of blood pressure is important for the primary prevention of cardiovascular disease (CVD).

Whilst there is an established link between depression and CVD, there have been mixed findings in the association between depression and HTN [2–6]. However, irrespective of whether patients with depression are more or less likely to develop hypertension, it is still important that they receive appropriate clinical care and management of their blood pressure for the prevention of CVD. Patients on long-term antidepressants having a National Health Service (NHS) health check in England were more likely to be identified as having HTN and to be prescribed antihypertensive therapy (AHT), however, this study did not examine blood pressure control [7]. Further, Byrd et al found that whilst high blood pressure was detected earlier in patients with depression and anxiety, recognition of HTN (by a recording of ICD9 code or prescription of AHT) was similar to patients without documented mental health disorders [8]. Another study using electronic health records (EHR) found that patients who had depression and/or anxiety were more likely to achieve control of blood pressure (<140/90 mmHg) and did so quicker than those without mental health disorders. Being female was also a significant predictor of faster blood pressure control, but whether there were differences in females with and without mental health disorders was not reported [9]. Notably, these studies did not look at both treatment and control of blood pressure within the depressed population as part of routine care.

Whilst the association between low socio-economic status and increased risk of cardiovascular disease and depression is well known, the impact on treatment and control of blood pressure is less clear, particularly in the depressed [10,11]. A study in older (60–79 years) adults found no association between socio-economic factors and blood pressure control, however this study did not investigate depression [12]. Likewise, evidence has indicated that those living in rural areas are less likely to have depression than those in urban environments [13]. In addition, rural general practices are less likely to prescribe blood pressure and lipid lowering drugs in patients with coronary heart disease or hypertension but whether there are differences in patients with depression is not known [14].

This study aimed to investigate whether patients with depression, free from atherosclerotic CVD at the time of assessment, who had a prior documented diagnosis of HTN were more or less likely to be on AHT medication and have documentation of controlled blood pressure within a year of depression diagnosis than patients without depression. Also, we wished to determine whether there were differences in

treatment and control of blood pressure in new incident diagnoses of hypertension in patients after a depression diagnosis compared to those without depression. Finally, we explored the potential impact of sex, socio-economic status (deprivation) and geographic location of residence on the associations for both new and historical HTN.

## Methods

### Availability of data and materials

The data used in this study are available in the Secure Anonymised Information Linkage (SAIL) Databank at Swansea University, Swansea, UK, but as restrictions apply, they are not publicly available. All proposals to use SAIL data are subject to review by an independent Information Governance Review Panel (IGRP). Before any data can be accessed, approval must be given by the IGRP. The IGRP carefully considers each project to ensure the proper and appropriate use of SAIL data. When access has been granted, it is gained through a privacy-protecting trusted research environment (TRE) and remote access system referred to as the SAIL Gateway. SAIL has established an application process to be followed by anyone who would like to access data via SAIL at https://saildatabank.com/data/apply-to-work-with-the-data/.

### Ethics approval and consent to participate

Approval for the use of anonymised data in this study, provisioned within the SAIL Databank, was granted by an independent Information Governance Review Panel (IGRP) under project 0800. The IGRP has a membership comprised of senior representatives from the British Medical Association (BMA), the National Research Ethics Service (NRES), Public Health Wales and Digital Health and Care Wales (DHCW). The usage of additional data was granted by each respective data owner. The SAIL Databank is compliant with General Data Protection Regulations (GDPR) and the UK Data Protection Act. Patient consent was not required by the IGRP as all data was anonymised prior to the study.

### Cohort

Patients who were part of the retrospective observational cohort study with a record of hypertension in their electronic health record (EHR) and at least 90 days of follow-up post hypertension diagnosis were included. The retrospective observational cohort study used individual-level linked anonymised routinely collected EHR data sources to identify patients who were free from atherosclerotic CVD, depression or severe mental illness, aged 18 years or over and had at least 1 year of data within SAIL prior to entry into the cohort. Patients entered the cohort on the 1st January 2010, or on the date of achievement of the inclusion criteria. Patients remained in the cohort until 31st December 2019, death or on leaving a SAIL providing General Practice.

Presence of hypertension, dyslipidaemia, ischaemic heart disease, chronic kidney disease (CKD) stage 4 + , chronic liver disease, dementia, cancer, depression, anxiety, severe mental illness, prescriptions for lipid lowering medication, antihypertensive therapy (AHT), antipsychotics, anxiolytics and antidepressant therapy were identified in primary care data. Secondary care data sources were also used to describe prior history of or contemporary myocardial infarction, peripheral vascular disease (PVD), heart failure, diabetes mellitus and ischaemic stroke. The Welsh Demographic Service Dataset (WDSD) was used to link other variables to the Lower-layer Super Output Area (LSOA) version 2001 of residence, and from this linkage to the area-based deprivation measure Welsh Index of Multiple Deprivation (WIMD) 2011. This is an indicator of socio-economic status and provides location (classified as rural and urban). The code used to generate the cohort is also available at (https://github.com/r-sum-1/SAIL0800-CVD-Depression.git).

### Depression characterisation

Welsh Longitudinal General Practice (WLGP) data were used to identify patients who had a record of any of the following: a diagnosis of depression or mixed anxiety and depression, anxiety, severe mental illness (including bipolar disorder,

schizophrenia and other psychotic illnesses), depressive symptoms, anxiety symptoms, prescriptions for antidepressants or anxiolytics. A list of diagnostic (Read) codes was created based on previous work from our group [15]. Patients were categorised as having depression during the study period if they met the following criteria (to be identified as depression forthwith), a diagnosis of depression or mixed anxiety and depression in their medical history, or a record of depressive symptoms together with a prescription of antidepressants within 6 months of record of the depressive symptoms. Depressive symptoms were included in the depression categorisation to reflect changes within coding behaviour as specified in John et al [15], with only those patients with both symptoms and a prescription for antidepressants being included in the depressed group. This approach has been validated through linkage to survey data by John et al [15]. Patients prescribed antidepressants but without a record of depression diagnosis or symptoms were not categorised as depressed.

### Hypertension identification, treatment and control

Read and ICD-10 codes were used to identify diagnoses of hypertension in EHRs. Hypertension identified prior to date of entry to the cohort for those without depression or prior to date of depression diagnosis for those with depression was categorised as prior diagnosis of hypertension. Hypertension identified post entry to cohort or depression diagnosis was categorised as new incident hypertension.

Prescriptions for AHT were identified. Patients who had a record for a new prescription for AHT within one year of new incident HTN diagnosis date or one year of entry to cohort for history of HTN were counted as on treatment.

Documented blood pressure readings were identified during the year post entry into the study for prior HTN and year post HTN diagnosis for new incident HTN. For patients with multiple readings the measurement taken closest to the 1 year time limit was used for assessing blood pressure control. Control was defined as having a blood pressure reading <140/90 mmHg.

### Statistical analysis

Variables are presented as mean (standard deviation [SD]) for continuous and frequency (percentage) for categorical variables. Comparisons between depressed and non-depressed groups were carried out using a two-sample t-test or chi square as appropriate, as were additional comparisons by sex, age, deprivation and location of residence. The percentage of data missing from WIMD was 1.6% and location of residence was 1.0%, no imputation was performed.

Logistic regression was used to investigate whether depression status was associated with the main outcomes of being on AHT within one year of cohort entry and whether blood pressure was controlled within one year of entry to cohort for history of HTN patients and prescription of AHT treatment within 1 year of HTN diagnosis and control of blood pressure within 1 year of diagnosis for new incident HTN. Models were adjusted for sex, age group (18–39, 40–59, 60–74 & 75+), deprivation, location of residence, history of diabetes, CKD, liver disease and cancer. Analyses for blood pressure control were also adjusted for AHT. For all models, all 2 and 3-way interactions between depression, sex and age were explored. Analyses were carried out using SPSS version 26 and R version 2024.24.0.

## Results

A total of 289,691 patients were included in the analysis (S1 Fig), 237,435 had a history of HTN and 52,256 had a record within their EHR during the study period. Table 1 contains the baseline characteristics for the two groups. Those with a history of HTN were older, had a greater proportion of patients with depression, history of diabetes and dyslipidaemia than the new incident HTN group.

### Blood pressure treatment and control in prior diagnosis hypertension

**Treatment.**  Of the 237,435 patients with a prior diagnosis of HTN, a greater proportion of those diagnosed with depression (depressed) were prescribed AHT within the first year of the study evaluation than non-depressed patients

**Table 1. Baseline characteristics for patients with a history of hypertension and new incident of hypertension by depression status.**

| | History HTN | | | New HTN | | |
|---|---|---|---|---|---|---|
| | **Non-depressed** | **Depressed** | **p** | **Non-depressed** | **Depressed** | **p** |
| n (%) | 218320 (91.9) | 19115 (8.1) | | 50854 (97.3) | 1402 (2.7) | |
| Age years mean (SD) | 65.3 (13.2) | 60.6 (15.3) | <0.001 | 63.6 (14.4) | 49.0 (14.8) | <0.001 |
| Characteristic n (%) | | | | | | |
| Female | 10041 (50.4) | 10489 (54.9) | <0.001 | 24662 (48.5) | 784 (55.9) | <0.001 |
| Deprivation index WIMD quintile* | | | <0.001 | | | <0.001 |
| 1 (most deprived) | 36230 (16.9) | 4595 (24.3) | | 7372 (14.8) | 380 (27.3) | |
| 2 | 42024 (19.6) | 4102 (21.7) | | 9765 (19.6) | 301 (21.6) | |
| 3 | 46819 (21.8) | 3758 (17.4) | | 9908 (19.9) | 272 (19.5) | |
| 4 | 45193 (21.0) | 3224 (17.0) | | 12663 (25.4) | 228 (16.4) | |
| 5 (least deprived) | 44642 (20.8) | 3235 (17.1) | | 10154 (20.4) | 211 (15.2) | |
| Location of residence** | | | <0.001 | | | <0.001 |
| Rural | 75578 (35.0) | 5466 (28.7) | | 20719 (41.2) | 372 (26.6) | |
| Urban | 140518 (65.0) | 13564 (71.3) | | 29537 (58.8) | 1028 (73.4) | |
| Past medical history n (%) | | | | | | |
| Dyslipidaemia | 41240 (18.9) | 3085 (16.1) | <0.001 | 2591 (5.1) | 129 (9.2) | <0.001 |
| Diabetes Mellitus | 36488 (16.7) | 3947 (20.6) | <0.001 | 3931 (7.7) | 167 (11.9) | <0.001 |
| Liver disease | 2821 (1.3) | 286 (1.5) | 0.017 | 255 (0.5) | 6 (0.4) | 0.70 |
| Chronic kidney disease 4+ | 2461 (1.1) | 292 (1.5) | <0.001 | 389 (0.8) | 14 (1.0) | 0.32 |
| Cancer | 27038 (12.4) | 1688 (8.8) | <0.001 | 4979 (9.8) | 108 (7.7) | 0.009 |

\* Total population History HTN n = 233822, New HTN n = 51254. ** Total population History HTN n = 235126, New HTN n = 51656.

(Table 2A). Depressed males and females with a prior diagnosis of HTN were more likely to be prescribed treatment than their non-depressed equivalents.

Non-depressed males with a prior diagnosis of HTN were more likely to be prescribed treatment than non-depressed females (males 77.0 vs females 74.1% p < 0.001) but there was no difference between depressed males and females (83.3 vs 82.3% (p = 0.08).

The proportion of patients prescribed treatment increased with age until 60–74, after which there was a decline in numbers prescribed AHT for both the depressed and non-depressed groups. Overall, a greater proportion of depressed patients were prescribed treatment than non-depressed in any given age group.

When considering treatment according to deprivation quintile, depressed patients with a prior diagnosis of HTN were more likely to be prescribed AHT than non-depressed patients in the equivalent quintile. In depressed patients, AHT prescription was similar across deprivation quintiles, whereas in non-depressed patients the least deprived were the most likely to receive AHT (Table 2A).

With respect to location of residence, prescription of AHT was similar in both rural and urban areas for depressed patients, whereas non-depressed patients in urban areas were more likely to be prescribed AHT (Table 2A).

In those with a prior diagnosis of HTN, depression was independently associated with being prescribed AHT (OR 95%CI 1.71, 1.64–1.78) as were age, sex, deprivation and location of residence in the multivariable analysis (Tables 3A and S2A). Exploring interactions between depression, age and sex, a significant depression*age effect was found, with the increased likelihood of AHT prescription most pronounced in the older age groups (Fig 1A).

**Blood pressure control.** Patients with a prior diagnosis of HTN, who had a valid blood pressure measurement recorded within the year following entry to the study were included in the analysis (181,174 [76.3%]). Of these 15,406

**Table 2. Number (%) of patients with a prior hypertension diagnosis, with and without depression, who were prescribed A) antihypertensive therapy and B) achieved blood pressure control, during the first year of inclusion in the study by sex, age, deprivation and location of residence.**

| | A)Prescription of antihypertensive therapy | | | Within ND | Within D | B)Blood pressure control | | | Within ND | Within D |
|---|---|---|---|---|---|---|---|---|---|---|
| | Non-depressed | Depressed | p | p | p | Non-depressed | Depression | p | p | p |
| **All** | 164989 (75.6) | 15819 (82.8) | <0.001 | | | 78436 (47.3) | 8756 (56.8) | <0.001 | | |
| **Sex** | | | | <0.001 | 0.081 | | | | <0.001 | <0.001 |
| Male | 83398 (77.0) | 7184 (83.3) | <0.001 | | | 38827 (46.8) | 3736 (54.9) | <0.001 | | |
| Female | 81591 (74.1) | 8635 (82.3) | <0.001 | | | 39609 (47.9) | 5020 (58.4) | <0.001 | | |
| **Age group** | | | | <0.001 | <0.001 | | | | <0.001 | <0.001 |
| 18-39 | 3844 (50.6) | 896 (55.3) | <0.001 | | | 2453 (52.6) | 686 (61.6) | <0.001 | | |
| 40-59 | 47101 (74.8) | 6685 (83.1) | <0.001 | | | 22954 (49.1) | 3504 (55.2) | <0.001 | | |
| 60-74 | 74403 (78.8) | 5081 (89.0) | <0.001 | | | 35103 (47.1) | 2738 (57.0) | <0.001 | | |
| 75+ | 39641 (74.3) | 3157 (84.5) | <0.001 | | | 17926 (44.9) | 1828 (58.3) | <0.001 | | |
| **Deprivation quintiles WIMD** | | | | <0.001 | 0.88 | | | | 0.042 | 0.89 |
| 1(most deprived) | 28267 (78.0) | 3800 (82.7) | <0.001 | | | 13323 (48.0) | 2110 (57.2) | <0.001 | | |
| 2 | 31601 (75.2) | 3386 (82.5) | <0.001 | | | 15050 (47.3) | 1887 (57.0) | <0.001 | | |
| 3 | 36025 (76.9) | 3130 (83.3) | <0.001 | | | 16980 (46.8) | 1336 (43.9) | <0.001 | | |
| 4 | 32477 (71.9) | 2656 (82.4) | <0.001 | | | 15592 (47.1) | 1466 (56.7) | <0.001 | | |
| 5 (least deprived) | 34175 (76.6) | 2679 (82.8) | <0.001 | | | 16336 (47.5) | 1497 (57.4) | <0.001 | | |
| **Location of residence** | | | | <0.001 | 0.73 | | | | <0.001 | 0.42 |
| Rural | 55884 (73.9) | 4532 (82.9) | <0.001 | | | 27043 (48.4) | 2528 (57.3) | <0.001 | | |
| Urban | 107508 (76.5) | 11218 (82.7) | <0.001 | | | 50629 (46.8) | 6187 (56.6) | <0.001 | | |

ND; non-depressed, D; depressed.

(8.5%) were diagnosed with depression. Those patients with a blood pressure measurement were older and more likely to have a history of dyslipidaemia and diabetes than those without a follow-up reading (S1 Table).

Patients with depression were slightly more likely to have a blood pressure assessment than non-depressed (80.6 vs 75.9% p < 0.001). A greater proportion of depressed patients had documentation of a controlled blood pressure (<140/90 mmHg) than non-depressed (depressed 56.8 vs non-depressed 47.3% p < 0.001, Table 2B).

Female patients were more likely to have a controlled blood pressure documented than males in both the depressed and non-depressed groups (Table 2B).

With regard to age, patients with depression were more likely to have documented blood pressure control in all age groups. For both non-depressed and depressed patients, those aged 18–39 years were most likely to have controlled blood pressure documented (Table 2B).

In depressed patients with a prior diagnosis of HTN documented blood pressure control was similar between all deprivation quintiles. For non-depressed patients those in the deprivation quintile 3 were the least likely to have controlled blood pressure and the most deprived group the most likely (Table 2B).

With regard to location of residence non-depressed patients living in rural areas were most likely to have documentation of controlled blood pressure. There were no differences in documentation of controlled blood pressure between urban and rural areas for depressed patients (Table 2B).

**Table 3. Main effects multivariable binary logistic regression estimates of the association of depression with A) being prescribed antihypertensive therapy and B) blood pressure control in patients with a prior diagnosis of hypertension.**

| A | OR | 95% C.I. | p | B | OR | 95% C.I. | p |
|---|---|---|---|---|---|---|---|
| Depression | 1.71 | 1.64 - 1.78 | <0.001 | Depression | 1.42 | 1.37 - 1.46 | <0.001 |
| Age group | | | <0.001 | Age group | | | <0.001 |
| 40-59 | 3.00 | 2.86 - 3.13 | <0.001 | 40-59 | 0.84 | 0.79 - 0.89 | <0.001 |
| 60-74 | 3.76 | 3.60 - 3.94 | <0.001 | 60-74 | 0.76 | 0.72 - 0.81 | <0.001 |
| 75+ | 3.02 | 2.88 - 3.16 | <0.001 | 75+ | 0.69 | 0.65 - 0.73 | <0.001 |
| Female | 0.86 | 0.84 - 0.88 | <0.001 | Female | 1.10 | 1.08 - 1.12 | <0.001 |
| Deprivation quintiles WIMD | | | <0.001 | Deprivation quintiles WIMD | | | 0.001 |
| 2 | 0.87 | 0.84 - 0.90 | <0.001 | 2 | 0.98 | 0.95 - 1.01 | 0.10 |
| 3 | 0.97 | 0.93 - 1.00 | 0.031 | 3 | 0.95 | 0.92 - 0.98 | 0.001 |
| 4 | 0.75 | 0.72 - 0.77 | <0.001 | 4 | 0.96 | 0.93 - 0.99 | 0.013 |
| 5 (least deprived) | 0.92 | 0.89 - 0.95 | <0.001 | 5 (least deprived) | 1.00 | 0.97 - 1.03 | 0.87 |
| Location of residence: Urban | 1.11 | 1.09 - 1.14 | <0.001 | Location of residence: Urban | 0.92 | 0.90 - 0.94 | <0.001 |
| History of diabetes | 1.33 | 1.29 - 1.37 | <0.001 | History of diabetes | 1.33 | 1.30 - 1.37 | <0.001 |
| History of chronic kidney disease | 1.11 | 1.01 - 1.22 | 0.025 | History of chronic kidney disease | 1.39 | 1.27 - 1.52 | <0.001 |
| History of dyslipidaemia | 1.44 | 1.40 - 1.48 | <0.001 | History of dyslipidaemia | 1.05 | 1.03 - 1.08 | <0.001 |
| History of liver disease | 0.74 | 0.68 - 0.80 | <0.001 | History of liver disease | 1.10 | 1.02 - 1.20 | 0.021 |
| History of cancer | 0.96 | 0.93 - 0.99 | 0.006 | History of cancer | 1.09 | 1.06 - 1.12 | <0.001 |
| Constant | 1.02 | | 0.57 | AHT within 1 year | 1.06 | 1.03 - 1.09 | <0.001 |
| | | | | Constant | 1.05 | | 0.11 |

In the multivariable analysis, depression was independently associated with having a controlled blood pressure documented within 1 year of entry to study in patients with a prior diagnosis of HTN (OR 95%CI 1.42, 1.37–1.46). Additionally, being female was associated with having documentation of controlled blood pressure, whilst increasing age was associated with decreasing blood pressure control (Tables 3B and S2B). Exploring interactions, there was a significant three-way interaction between depression, sex and age group. For males, the difference between depressed and non-depressed blood pressure control rates was small in the young, and tended to increase consistently with age. In contrast for females, there was a higher control rate for the depressed group at all ages, notably in the young population (Fig 1B).

### Blood pressure treatment and control in those with new incident hypertension

**Treatment.** Of the 52,256 patients with a new diagnosis of HTN recorded in their EHR, 1,402 (2.7%) had depression. Depressed patients were significantly younger at time of HTN diagnosis than non-depressed patients (depressed 49.0 [14.8] vs non-depressed 63.6 [14.4] yrs p<0.001) and a greater proportion were female (depressed 55.9 vs non-depressed 48.5%). Depressed patients were also more likely to have been prescribed AHT within 1 year of recorded HTN diagnosis (depressed 63.8% vs non-depressed 31.6% p<0.001).

There was no difference in the proportion of male and female depressed patients prescribed AHT within a year of diagnosis, whereas non-depressed males were more likely to be prescribed AHT than non-depressed females (Table 4A).

The proportion of depressed patients prescribed AHT within 1 year of HTN diagnosis increased with age except beyond 75 years where there was a fall in numbers prescribed AHT. For non-depressed patients, similar proportions were treated in the 18–39 and 40–59 age groups, followed by a decrease in the proportion of patients prescribed AHT within 1 year (Table 4A).

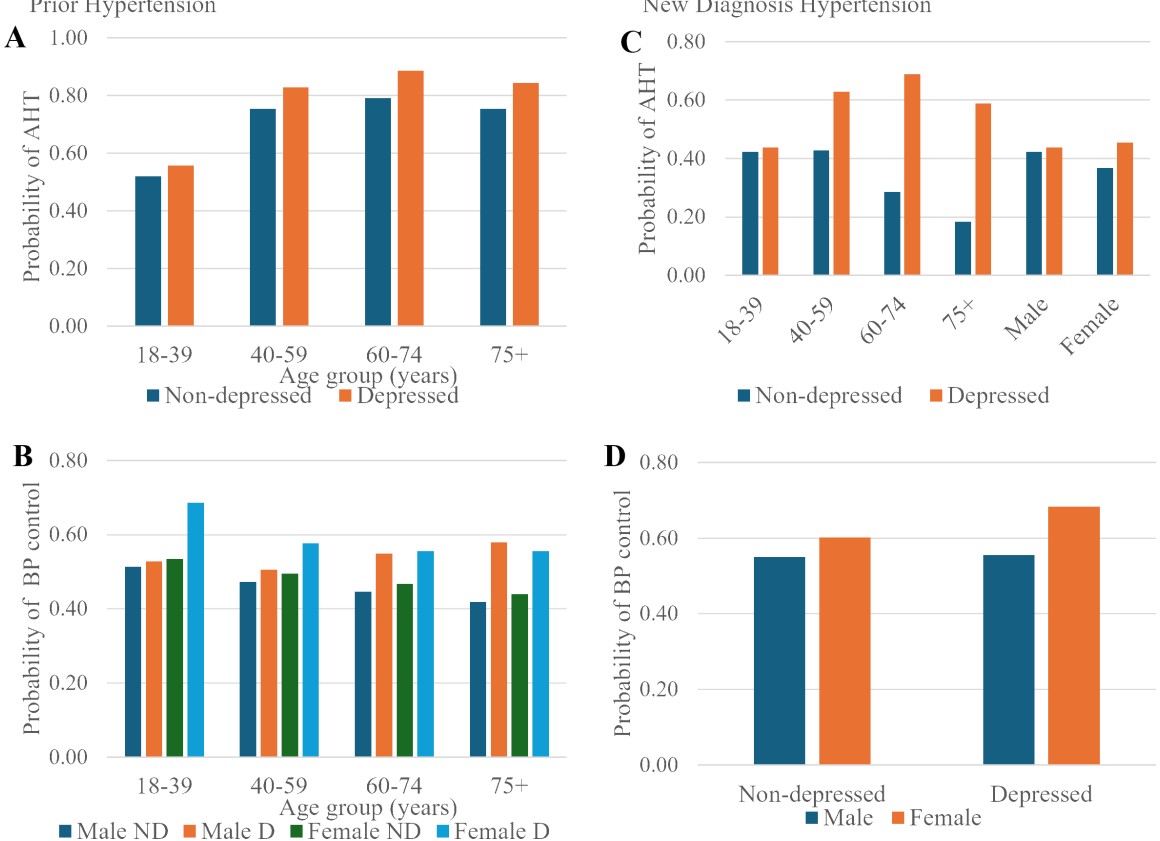

**Fig 1. Model estimates, showing significant interactions between depression, sex and age, for the prescription of antihypertensive therapy (left panel) and blood pressure control (right panel).** A and B show prior hypertension group, C and D show new diagnosis of hypertension group. All estimates were adjusted for sex, age group, WIMD, location of residence, history of diabetes, dyslipidaemia, liver disease, chronic kidney disease and cancer plus prescription of antihypertensive therapy for B and D.

Considering deprivation status, patients with depression were more likely to be prescribed AHT than non-depressed patients at all quintiles of deprivation. In the depressed group, prescribing of AHT was similar between the quintiles of deprivation (Table 4A). For the non-depressed patients those in the most deprived quintiles were the most likely to be prescribed AHT (Table 4A).

Exploring differences due to location of residence, patients in the non-depressed group who lived in urban areas were the most likely to be prescribed AHT. Within the depressed group, a similar proportion were prescribed AHT in both urban and rural areas (Table 4A).

Depression was independently associated with being prescribed AHT within 1 year of a HTN diagnosis recording in EHR (OR 95%CI 2.67, 2.38–2.99) in multivariable regression. Female sex was associated with a lower likelihood of AHT prescription. Older patients were also less likely to be prescribed AHT (Tables 5A and S4A). Significant interactions were depression*age, which showed an increasing relative likelihood of AHT prescription in the depression groups with ageing; and depression*sex, which indicated that the increased likelihood of AHT prescription in the depression group was small for males and much more pronounced for females (Fig 1C).

**Blood pressure control.** Of those patients with a new HTN diagnosis recorded in their EHR, 19,327 (37.0%) who had a valid blood pressure measurement in the year following HTN diagnosis were included in this analysis. Depressed

**Table 4. Number (%) of patients with and without depression A) prescribed antihypertensive therapy and B) who had controlled blood pressure, within 1 year of HTN diagnosis by sex, age, deprivation and location of residence.**

| | A) Prescription of antihypertensive therapy | | | | | B) Blood pressure control | | | | |
| --- | --- | --- | --- | --- | --- | --- | --- | --- | --- | --- |
| | | | | Within ND | Within D | | | | Within ND | Within D |
| | Non-depressed | Depressed | p | p | p | Non-depressed | Depressed | p | p | p |
| **All** | 16063 (31.6) | 895 (63.8) | <0.001 | | | 9618 (52.6) | 617 (59.2) | <0.001 | | |
| **Sex** | | | | <0.001 | 0.96 | | | | <0.001 | <0.001 |
| Male | 9224 (35.2) | 395 (63.9) | <0.001 | | | 5133 (50.3) | 230 (51.8) | 0.54 | | |
| Female | 6839 (27.7) | 500 (63.8) | <0.001 | | | 4485 (55.5) | 387 (64.7) | <0.001 | | |
| **Age** | | | | <0.001 | <0.001 | | | | <0.001 | 0.14 |
| 18-39 | 1276 (42.4) | 189 (48.6) | 0.021 | | | 922 (57.1) | 169 (63.3) | 0.056 | | |
| 40-59 | 7194 (42.9) | 479 (68.1) | <0.001 | | | 4065 (50.6) | 298 (55.9) | 0.017 | | |
| 60-74 | 5619 (28.2) | 182 (74.9) | <0.001 | | | 3339 (52.5) | 117 (60.9) | 0.020 | | |
| 75+ | 1974 (17.7) | 45 (67.2) | <0.001 | | | 1292 (57.0) | 33 (66.0) | 0.20 | | |
| **Deprivation quintiles WIMD** | | | | <0.001 | 0.30 | | | | 0.001 | 0.59 |
| 1 (most deprived) | 3006 (40.8) | 228 (60.0) | <0.001 | | | 1639 (50.2) | 154 (57.2) | 0.027 | | |
| 2 | 3177 (32.5) | 194 (64.5) | <0.001 | | | 1935 (52.0) | 145 (62.0) | 0.003 | | |
| 3 | 3332 (33.6) | 180 (66.2) | <0.001 | | | 2021 (52.2) | 119 (57.8) | 0.12 | | |
| 4 | 3029 (23.9) | 141 (61.8) | <0.001 | | | 1882 (54.0) | 94 (56.6) | 0.51 | | |
| 5 (least deprived) | 3271 (32.2) | 143 (67.8) | <0.001 | | | 2009 (54.8) | 101 (63.1) | 0.038 | | |
| **Location of residence** | | | | <0.001 | 0.09 | | | | 0.84 | 0.15 |
| Rural | 5198 (25.1) | 224 (60.2) | <0.001 | | | 3079 (52.7) | 166 (64.2) | 0.001 | | |
| Urban | 10739 (36.4) | 669 (65.1) | <0.001 | | | 6467 (52.6) | 449 (57.9) | 0.004 | | |

patients accounted for 5.4% of this population. Those patients with a blood pressure measurement were younger, male and more likely to have dyslipidaemia and diabetes than those without a follow-up assessment (S3 Table).

Those with depression were more likely to have a blood pressure assessment than non-depressed (depressed 74.3% vs non-depressed 36.0% p<0.001). Of those with a valid blood pressure reading, patients with depression were more likely to have controlled blood pressure documented within 1 year of HTN diagnosis than non-depressed patients (59.2 vs 52.6% p<0.001).

Female patients were more likely to have documentation of controlled blood pressure than male in both the depressed and non-depressed groups with the greatest proportion of control in the depressed females (Table 4B). There was no difference in blood pressure control documentation between depressed and non-depressed males.

When considering documentation of blood pressure control by age group, the depressed patients were more likely to have controlled blood pressure documented than the non-depressed in the 40–59 and 60–74 age groups (Table 4B).

Considering deprivation status, in depressed patients, documentation of controlled blood pressure was similar between the quintiles of deprivation. In non-depressed patients, those in the most deprived quintiles were the least likely to have documented blood pressure control (Table 4B).

With regard to location of residence, depressed patients in both urban and rural areas, were more likely to have documentation of controlled blood pressure than non-depressed. However, there were no differences in documentation of control between depressed patients in urban and rural areas and likewise for the non-depressed patients (Table 4B).

**Table 5. Main effects multivariable binary logistic estimates for the association of depression with A) being prescribed antihypertensive therapy and B) blood pressure control in patients with new incident hypertension.**

| A | OR | 95% C.I. | p | B | OR | 95% C.I. | p |
|---|---|---|---|---|---|---|---|
| Depression | 2.67 | 2.38 - 3.00 | <0.001 | Depression | 1.23 | 1.08 - 1.40 | 0.002 |
| Age group | | | <0.001 | Age group | | | <0.001 |
| 40-59 | 1.12 | 1.03 - 1.21 | 0.006 | 40-59 | 0.76 | 0.69 - 0.85 | <0.001 |
| 60-74 | 0.6 | 0.55 - 0.65 | <0.001 | 60-74 | 0.76 | 0.69 - 0.85 | <0.001 |
| 75+ | 0.34 | 0.31 - 0.37 | <0.001 | 75+ | 0.86 | 0.76 - 0.98 | 0.024 |
| Female | 0.80 | 0.77 - 0.83 | <0.001 | Female | 1.26 | 1.19 - 1.33 | <0.001 |
| Deprivation quintiles WIMD | | | <0.001 | Deprivation quintiles WIMD | | | <0.001 |
| 2 | 0.80 | 0.75 - 0.86 | <0.001 | 2 | 1.08 | 0.98 - 1.18 | 0.12 |
| 3 | 0.93 | 0.87 - 0.99 | 0.029 | 3 | 1.07 | 0.98 - 1.18 | 0.13 |
| 4 | 0.62 | 0.58 - 0.67 | <0.001 | 4 | 1.15 | 1.05 - 1.27 | 0.004 |
| 5 (least deprived) | 0.85 | 0.80 - 0.91 | <0.001 | 5 (least deprived) | 1.22 | 1.11 - 1.34 | <0.001 |
| Location of residence: Urban | 1.42 | 1.36 - 1.48 | <0.001 | Location of residence: Urban | 0.99 | 0.93 - 1.06 | 0.79 |
| History of diabetes | 1.37 | 1.28 - 1.47 | <0.001 | History of diabetes | 1.82 | 1.65 - 2.01 | <0.001 |
| History of chronic kidney disease | 1.02 | 0.78 - 1.34 | 0.89 | History of chronic kidney disease | 2.65 | 1.61 −4.39 | <0.001 |
| History of dyslipidaemia | 2.23 | 2.05 - 2.42 | <0.001 | History of dyslipidaemia | 1.29 | 1.15 - 1.43 | <0.001 |
| History of liver disease | 1.05 | 0.84 - 1.30 | 0.69 | History of liver disease | 1.56 | 1.12 - 2.19 | 0.009 |
| History of cancer | 1.25 | 1.17 - 1.34 | <0.001 | History of cancer | 1.19 | 1.08 - 1.32 | <0.001 |
| Constant | 0.67 | | <0.001 | AHT within 1 year | 0.82 | 0.76 - 0.88 | <0.001 |
| | | | | Constant | 1.22 | | 0.005 |

Depression was independently associated with having a documented controlled blood pressure within 1 year of a record of HTN diagnosis in EHR in multivariate analysis (OR 95%CI 1.23, 1.08–1.40). Age, sex, WIMD and location of residence were all associated with control (Tables 5B and S4B). A significant interaction was found between depression and sex, indicating that the greater likelihood of blood pressure control found for the depressed group was largely confined to the female population (Fig 1D).

## Discussion

In this study exploring the treatment and control of blood pressure in patients with a prior diagnosis of or new incident HTN and with and without depression in the primary care environment, we found that overall, those with depression were more likely to be prescribed AHT and have their blood pressure assessed. With regards to control, women with depression were generally more likely to have a blood pressure <140/90 mmHg within the year of diagnosis of depression or subsequent diagnosis of HTN post depression diagnosis.

The finding that depressed patients were more likely to be prescribed AHT does agree with Garriga et al who found that health check attendees on long term antidepressants were more likely to be prescribed AHT within 12 months than those not on antidepressants [7]. It was notable in both the populations with a prior diagnosis of and new incident HTN that women were the least likely to be prescribed AHT in both unadjusted and adjusted analyses. Although the 2-way interaction between depression*sex in the new incident HTN did indicate that depressed women were most likely to be treated and non-depressed females the least likely. Previous work such as Kiss et al found no sex differences in AHT prescribing in patients free from CVD with elevated blood pressure >140/90 mmHg [16]. Whilst studies including patients with CVD did find that women were less likely to be prescribed angiotensinogen converting inhibitors or calcium channel blockers but more likely to be prescribed diuretics [17,18], these studies have not looked at sex differences in those with depression.

Women with a prior diagnosis of HTN or with a new HTN diagnosis during the study period were more likely to have controlled blood pressure within one year, particularly in those with depression, which was consistently associated with more effective blood pressure control. Similarly, in an analysis of baseline data from the PREDIMED study looking at blood pressure in high CV risk patients with and without depression, those with depression were more likely to be prescribed AHT and have controlled blood pressure [19]. Furthermore, Ho et al., found in their study that patients with depression/anxiety had faster rates of control to those without with females achieving target blood pressure levels more quickly in general than males, although they did not look specifically at differences between depressed and non-depressed patients according to sex [9].

One important potential confounder when considering differences in blood pressure control according to sex is that females may be diagnosed earlier and with lower blood pressures than men, albeit still in the hypertensive range, at the time of HTN diagnosis. This may be due to females being more likely to have their blood pressure assessed and so may have HTN identified at an earlier less severe level and also may be more likely to adopt lifestyle changes successfully for reducing blood pressure independently from medication. For example, in the current study the depressed patients were significantly younger than the non-depressed. Another proposed reason for greater levels of control in females is that they may not require as high dosage or numbers of AHT medication(s) to achieve controlled blood pressure as males [16,20]. However, in the current study it was not possible to look at prescribed doses or scheduling of AHT to investigate whether there were differences between the sexes with regard to these parameters.

The finding that patients who lived in urban areas were more likely to be prescribed AHT for both prior and new incident HTN agrees with previous work [14]. Conversely, those with prior HTN living in urban areas were less likely to achieve control of blood pressure following multivariate analyses. However, these differences seem to be driven by lower levels of control in the non-depressed group. For whilst there were a greater proportion of depressed patients in urban areas there was no difference in likelihood of being prescribed AHT or achieving blood pressure control. Similarly with the deprivation index there was no clear trend within the depressed group in the likelihood of being prescribed AHT or achieving blood pressure control, indicating an equitable outcome in blood pressure control across the socioeconomic groups.

Prescription of AHT in patients with new incident HTN was lower than in those with a prior HTN diagnosis, particularly in the non-depressed group. United Kingdom based National Institute for Health and Care Excellence (NICE) guidelines, that were in practice during the study period, state that AHT should be offered to those patients with persistent stage 2 HTN [21]. Patients in this study were free from atherosclerotic CVD prior to entry into the cohort, so likely to be at lower cardiovascular risk and consequently may not have the requisite blood pressure level for being prescribed AHT according to NICE.

In this study the target of <140/90 mmHg was used for control of blood pressure. Recent European Society of Cardiology guidelines for elevated blood pressure and hypertension control now recommend a target of 120–129 mmHg for systolic blood pressure in treated patients (except in those ≥85 years or with clinically significant moderate to severe frailty issues) [22]. Applying this updated target in our analyses would have likely further reduced the proportion of patients found to have achieved blood pressure control and indicates the need for greater efforts in this area given the already relatively suboptimal levels of control in both the depressed and non-depressed populations. New recommendations have also included annual follow-up once blood pressure is stable and controlled. It is worth noting that whilst 76.3% of patients with a prior diagnosis of HTN had a blood pressure assessment within a year of entering the cohort there remains a very large number who did not receive a check. In the new incident HTN group there was no follow-up blood pressure in the majority of patients, with just over a third (37.0%) of patients being reassessed within the first year following diagnosis. However, patients with depression were more likely to be assessed, which may reflect the opportunity taken due to increased interaction with healthcare professionals in those with depression.

## Limitations/strengths

There are a number of limitations to this study. Blood pressure at time of new incident HTN diagnosis was not included in the study as a large proportion of patients did not have a measurement documented in their EHR at the time of, or within a year of the HTN diagnosis. Additionally, as timing of the blood pressure measurements varied within the population (up to a year) this may have added a bias to the results. This analysis was undertaken to look at early control of blood pressure in hypertensive patients following an initial diagnosis of HTN and/or depression. We have not addressed the relationship between long term control and clinical outcomes which would be the subject of further, more complex analysis. As with other studies that use real world observational EHR data it was not collected for research purposes, therefore there may be errors within the data that were or were not entered into the relevant datasets.

Differences in prescribing for and control of HTN may be influenced by consultation rate with health care professionals, which has not been included in this study. It has not been possible to explore differences within the population due to ethnicity due to a large proportion of missing data. However, the recent creation of a national ethnicity spine using data held within SAIL may enable this in future work [23]. Adherence to AHT is important in the control of blood pressure, however, it was not possible to look at this key variable or dispensing data as a proxy as only prescribing data is available in the EHR. This study has focused on classical risk factors for cardiovascular disease, future work interrogating sex differences within the general population and those with depression could include sex specific risk factors such as gestational hypertension and other pregnancy complications.

This data is based primary care data within the Welsh population within a free to access National healthcare system. Whilst not generalisable to all health care systems the majority of primary prevention occurs in primary care and so therefore provides an overview of care for the majority of the primary prevention population. Analyses of real world healthcare outcomes from United Kingdom datasets are very often representative of other Western populations but would clearly require replication in other populations of interest.

In conclusion patients with depression identified in primary care are more likely to be prescribed AHT and have documentation of effective blood pressure control than those without depression. These patterns were observed both in those with HTN identified before and following the depression diagnosis. Of these patients, depressed women were the most likely to achieve controlled blood pressure. In addition, depressed patients were likely to have similar levels of AHT prescribing and blood pressure control irrespective of their deprivation status or location of residence indicating achievement of therapeutic equity in these patient groups. However, it should be acknowledged that inequities exist in development of both clinical cardiovascular disease and depression, particularly in lower socioeconomic groups. Plus, there is more work to do for non-depressed patients in these areas who did not achieve the same levels of blood pressure control. More work is required to achieve greater levels of blood pressure control in patients with and without depression for the primary prevention of cardiovascular disease.

## Supporting information

**S1 Fig. Flow chart showing cohort selection. aCVD, atherosclerotic cardiovascular disease; GP, General Practice data; MH, mental health conditions; HTN hypertension.**
(TIF)

**S1 Table. Characteristics of patients with a prior diagnosis of hypertension with and without a valid follow-up blood pressure assessment.**
(DOCX)

**S2 Table. Unadjusted binary logistic regression estimates of predictors of A) being prescribed antihypertensive therapy and B) blood pressure control in patients with a prior diagnosis of hypertension.**
(DOCX)

**S3 Table. Characteristics of patients with new incident hypertension with and without a valid follow-up blood pressure assessment.**
(DOCX)

**S4 Table. Unadjusted binary logistic estimates of predictors of A) being prescribed antihypertensive therapy and B) blood pressure control in patients with new incident hypertension.**
(DOCX)

## Acknowledgments

This study makes use of anonymised data held in the Secure Anonymised Information Linkage (SAIL) Databank. We would like to acknowledge all the data providers who make anonymised data available for research. Additionally, we would like to thank our public and patient group members for their contributions to the study.

## Author contributions

**Conceptualization:** Elizabeth A Ellins, Julian P. Halcox.

**Data curation:** Richard Summers, Carla White.

**Formal analysis:** Elizabeth A Ellins, Richard Summers, Carla White, Michael B. Gravenor, Julian P. Halcox.

**Funding acquisition:** Elizabeth A Ellins, Ann John, David PJ. Osborn, Keith Lloyd, Ashley Akbari, Michael B. Gravenor, Julian P. Halcox.

**Methodology:** Ann John, David PJ. Osborn, Keith Lloyd, Ashley Akbari, Michael B. Gravenor, Julian P. Halcox.

**Project administration:** Elizabeth A Ellins.

**Supervision:** Julian P. Halcox.

**Writing – original draft:** Elizabeth A Ellins.

**Writing – review & editing:** Richard Summers, Carla White, Ann John, David PJ. Osborn, Keith Lloyd, Ashley Akbari, Michael B. Gravenor, Julian P. Halcox.

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
