## [Decision Letter · Decision Letter 0]

Treatment and control of blood pressure in Welsh patients with and without depression: a study of whole-population electronic health records

PONE-D-25-14314

Dear Dr. Ellins,

We’re pleased to inform you that your manuscript has been judged scientifically suitable for publication and will be formally accepted for publication once it meets all outstanding technical requirements.

Kind regards,

Swapnil Parve, M.D., Ph.D.

Academic Editor

PLOS ONE

Additional Editor Comments (optional):

Dear Authors, please consider the reviewer #2 comment: One of the reason behind the significant finding of early initiation of AHT and better BP control in the depressed group could be white coat effect in this population. There were no data on target organ damage like LVH in echocardiography in depressed Vs non-depressed population or inclusion of 24 hour Ambulatory BP monitoring.If data are available, can be included.

If you do not have anything to add, we may proceed to publication as this does not impact the strength of manuscript.

Reviewers' comments:

Reviewer's Responses to Questions

**Comments to the Author**

1. Is the manuscript technically sound, and do the data support the conclusions?

Reviewer #1: Yes

Reviewer #2: Yes

2. Has the statistical analysis been performed appropriately and rigorously? 

Reviewer #1: Yes

Reviewer #2: Yes

3. Have the authors made all data underlying the findings in their manuscript fully available?

Reviewer #1: Yes

Reviewer #2: Yes

4. Is the manuscript presented in an intelligible fashion and written in standard English?

Reviewer #1: Yes

Reviewer #2: Yes

5. Review Comments to the Author

Reviewer #1: I read with interest the manuscript authored by Elizabeth Ellins titled 'Treatment and control of blood pressure in Welsh patients with and without depression:a study of whole-population electronic health records.'

The research from analysis of electronic records provides details on a large community cohort. The authors have presented the methodology and results in detail and the manuscripot is well written.

I congratulate the authors for their efforts.

Reviewer #2: One of the reason behind the significant finding of early initiation of AHT and better BP control in the depressed group could be white coat effect in this population. There were no data on target organ damage like LVH in echocardiography in depressed Vs non-depressed population or inclusion of 24 hour Ambulatory BP monitoring.If data are available, can be included.

6. PLOS authors have the option to publish the peer review history of their article (what does this mean? ). If published, this will include your full peer review and any attached files.

**Do you want your identity to be public for this peer review?** For information about this choice, including consent withdrawal, please see our Privacy Policy .

Reviewer #1: **Yes: ** Jaideep C Menon

Reviewer #2: No

---

## [Editor Report · Acceptance letter]

PONE-D-25-14314

PLOS ONE

Dear Dr. Ellins,

I'm pleased to inform you that your manuscript has been deemed suitable for publication in PLOS ONE. Congratulations! Your manuscript is now being handed over to our production team.

Kind regards,

on behalf of

Dr. Swapnil Parve

Academic Editor

PLOS ONE